# Impact of COVID-19 on In-Patient and Out-Patient services in Bangladesh

**Ridwana Maher Manna**[1]*, **Md Hafizur Rahman**[1,2], **Tasnu Ara**[1], **Nasimul Ghani Usmani**[1], **K. M. Tanvir**[1], **M. Sabbir Haider**[3], **Ema Akter**[1], **Mohammad Sohel Shomik**[4], **Aniqa Tasnim Hossain**[1]

1 Maternal and Child Health Division, International Centre for Diarrhoeal Disease, Bangladesh (icddr,b), Dhaka , Bangladesh, 2 Biological Science Division, The University of Chicago, Chicago, Illinois, United States of America, 3 Institute of Epidemiology Disease Control and Research IEDCR, Dhaka, Bangladesh, 4 Nutrition and Clinical Services Division, International Centre for Diarrhoeal Disease, Bangladesh (icddr,b), Dhaka, Bangladesh

* ridwana.manna@icddrb.org

## Abstract

### Introduction

The global Coronavirus disease (COVID-19) pandemic disrupted healthcare systems, reducing access to medical services. In Bangladesh, strict lockdowns, healthcare worker shortages, and resource diversion further strained the system. Despite these challenges, the impact on inpatient and outpatient service utilisation in Bangladesh remains unaddressed. This study explored the levels of inpatient admissions and outpatient visits in public healthcare facilities before and during COVID-19 pandemic in Bangladesh.

### Methods

We conducted a cross-sectional secondary analysis of inpatient and outpatient data from all public hospitals collected via District Health Information System, version 2 (DHIS2) from January 2017 to June 2021. Using 2017-2019 as the baseline, we analysed healthcare utilisation indicators (outpatient visits and inpatient admissions) with descriptive and segmented Poisson regression to assess the impact of COVID-19 in 2020 and 2021.

### Results

In 2020, outpatient visits and inpatient admissions significantly declined to 34.1 million and 37.5 million, respectively, from 47.6 million and 56.2 million in 2019. Segmented regression analysis confirmed these drops, especially in Dhaka (IRR = 0.62, p<0.001) and Barisal (IRR = 0.69, p<0.002) for outpatient visits, and in Dhaka (IRR = 0.64, p<0.000) and Khulna (IRR = 0.70, p<0.000) for inpatient admissions. In 2021, most divisions saw an increase in outpatient visit and inpatient admission numbers, with the lowest rebound in Sylhet.

### Conclusion

The COVID-19 pandemic significantly reduced Outpatient Department (OPD) visits and Inpatient Department (IPD) admissions in Bangladesh in 2020, with partial recovery in

**Data availability statement:** Data for this analysis were sourced from the DHIS2 Bangladesh database, dedicated to health-related indicators for routine facility monitoring and evaluation of public healthcare facilities. This data is owned by the government of Bangladesh and only available with proper request to the government. For data inquiries, please contact Dr. Md Toufiq Hassan Shawon, Medical Officer at the Management Information System (MIS), Directorate General of Health Services (DGHS), at sawontheboss4@gmail.com. Dr. Shawon is responsible for managing and overseeing data access for this study.

**Funding:** The author(s) received no specific funding for this work.

**Competing interests:** The authors have declared that no competing interests exist.

2021. To ensure sustained access to care, it is crucial to strengthen healthcare facilities and equip healthcare providers to be prepared for future pandemics or emergencies.

## 1. Introduction

The Coronavirus disease (COVID-19), first identified in late 2019 in Wuhan, China, rapidly evolved into a global pandemic affecting 216 countries, resulting in over 770 million infections and 6 million deaths by September 24, 2023 [1]. The pathogen was identified by January 8, 2020, and by March 16, over 170,000 cases and more than 6,500 deaths had been confirmed worldwide, prompting large-scale lockdowns across nearly all nations to stem the virus spread and 'flatten the curve' [2]. These lockdowns had profound impacts across multiple sectors, disrupting food security, the global economy, education, tourism, hospitality, and healthcare, while also exacerbating gender-based violence, mental health issues, and environmental pollution [3].

The impact of the lockdown and high rates of COVID-19 infection had far-reaching effects on the people in different strata of life, including food security, global economy, education, tourism, hospitality, sports and leisure, gender relation, domestic violence/abuse, mental health, and environmental air pollution as well as reduced the availability of in-person medical services, limited access to healthcare, and healthcare service utilisation [3].

The healthcare sector, in particular, faced severe challenges. The fear of infection and unsafe working conditions led to significant stress among healthcare professionals (HCPs), contributing to reduced workforce availability and limiting healthcare service access and utilisation [4]. Studies from Uganda and Switzerland reflect this trend, with noticeable declines inpatient admissions and shifts in the patterns of healthcare usage during the pandemic [5,6].

In Bangladesh, the government reported its first COVID-19 case on March 8, 2020, followed by a stringent lockdown from March 17 to May 30, 2020 [7]. This period and subsequent pandemic wave triggered additional lockdowns throughout 2020 and 2021, significantly restricting mobility and impacting healthcare access [8]. The pandemic's toll was particularly harsh on healthcare workers, with 73 fatalities reported among professionals by August 9, 2020, representing approximately 10% of total infections early in the outbreak [9]. This high infection rate among healthcare workers highlighted the vulnerabilities within the healthcare system.

Amid these challenges, healthcare facilities in Bangladesh were compelled to adopt stringent infection prevention and control measures and triage systems to manage the dual burden of routine care and COVID-19 cases, which strained resources and led to further disruptions in service delivery [10]. Media reports and healthcare studies indicated a sharp decline in both inpatient and outpatient services, with the major divisions of Dhaka and Chattogram experiencing the highest infection and mortality rates, while Mymensingh reported the lowest [11–13]. For future pandemic preparedness, it is crucial to understand the systemic vulnerabilities and the adaptive responses of healthcare systems to ensure resilience and improved health outcomes in the face of global health crises. Therefore, this study explored the level of in-patient (IPD) admissions and out-patient (OPD) visits in public healthcare facilities before and during COVID-19 pandemic in Bangladesh.

## 2. Materials and methods

### 2.1. Study design

This study employed a cross-sectional secondary analysis of routine healthcare data to assess the impact of COVID-19 on health services utilisation indicators in Bangladesh.

## 2.2. Sources of data

The data used in this study were sourced from the District Health Information System, version 2 (DHIS2), which collects aggregated data from public healthcare facilities in Bangladesh [14]. The data included inpatient admission and outpatient visits from all 64 district hospitals across all eight admirative division of Bangladesh [15]. Health facilities record patient data at the facilities using paper-based or electronic systems. These data are then digitalised and uploaded to the DHIS2 platform by a designated facility staff in every facility. The uploaded data flows from the health facilities to district health offices, where it is reviewed, validated, and then sent to the national level [14].

DHIS2 employs a robust data validation process, including routine data check, cross-verification, and quality control in different administrative levels to ensure the accuracy and completeness of the data [14]. Although no direct methods such as surveys or interviews were employed in this study, the data was considered reliable due to the rigorous process DHIS2 implement for the quality assurance [16]. These processes include standardised data entry protocols at the facility level, where healthcare staff input data on outpatient visits and inpatient admissions using electronic or paper-based forms [17]. After entry, data undergoes multiple levels of review, including validation checks at district health offices, where errors or inconsistencies are flagged and corrected. Additionally, the DHIS2 platform perform automated data quality checks to ensure completeness and consistency before the data is consolidated at the national level [18].

## 2.3. Indicators

Two key indicators analysed were outpatient department (OPD) and inpatient department (IPD) totals. It is important to note that the data from DHIS2 is primarily aggregated from public healthcare facilities, and as such, certain departments or specialties within hospitals may not have been individually captured in this dataset. Additionally, the exclusion of data from private healthcare facilities limits the scope of the analysis to the public sector. This aggregated nature of the data restricts our ability to assess the impact of the pandemic on specific medical departments or services within hospitals, which may have been differently affected by COVID-19.

## 2.4. Ethical considerations

This study used anonymised data from the DHIS2 Bangladesh database, with access granted by the Directorate General of Health Services (DGHS). Data were collected in aggregate form to maintain confidentiality and privacy.

## 2.5. Data analysis

This study examined the annual trend of OPD visits and IPD admissions to determine changes in healthcare utilisation over time. Additionally, monthly trends were assessed to determine when COVID-19 had the biggest effects. We used segmented Poisson regression to quantify disruptions in monthly utilisation trends. We applied segmented regression, a statistical method that is frequently employed in interrupted time series (ITS) studies. To examine the count data, segmented regression with a Poisson distribution was used. The incidence rate ratios (IRR) are reported to explain the impact of COVID-19 (2020 and 2021) on our selected outcome variables compared with the pre-COVID period (from 2017 to 2019). We tested the autocorrelation function (ACF) using the Durbin–Watson statistic. The Dickey-Fuller unit root test was used to regulate seasonality in the model [19]. Stata version 15 was used to measure the monthly rate of change and trends.

## 3. Results

### 3.1. Overall OPD & IPD

Fig 1 shows the overall estimates of OPD visits and IPD admissions in Bangladesh over the five years period from 2017-2021. The number of OPD visits fell in 2020 (34.1 million) compared to 2019 but showed an upward trend, reaching the targeted predicted value in 2021. The number of IPD admissions likewise dropped during 2020 (37.5 million) compared to previous year (2019) which later increased in 2021 though remained comparatively low from the targeted predicted value.

### 3.2. OPD

Fig 2a depicts annual changes in the number of OPD visits from 2017 to 2021 for eight administrative divisions of Bangladesh. An upward trend of OPD visits was observed from 2017 to 2019 in all divisions except for Sylhet where slightly decrease was seen in 2018. In 2020, the number of OPD visits were dropped in all divisions compared to the pre COVID-19 years (2017-2019). Sudden decrease in the number of OPD visits was observed in Dhaka division from 10.7 million (2019) to 7.4 million (2020). Almost all the divisions countered an increasing trend in the number of OPD visits in 2021. The largest rebound after COVID-19 was seen in Barisal division (1.7 million in 2020 to 10.0 million in 2021) which was beyond the predicted value and the lowest rebound was noticed in Sylhet division (2.3 million in 2020 to 3.2 million in 2021).

### 3.3. IPD

Fig 2b shows the yearly changes in divisional IPD admissions within a five-year time period (2017-2021). The number of IPD admissions during the pre COVID-19 period (2017-2019) showed a small increase in all eight divisions except for Sylhet division where there was a decline in 2018. Throughout 2020, number of IPD admissions declined immensely in all divisions compared to the previous years (2017-2019), where a notable decrease was recorded in Dhaka division (1.1 million in 2019 to 0.68 million in 2020). A rebound was observed during 2021 in all divisions compared to the number of IPD admissions in 2020 but did not reach the predicted values. The largest bounce back was seen in Dhaka division (0.68 million in 2020 to 0.86 million in 2021) and the lowest rebound was remarked in Sylhet division (0.23 million in 2020 to 0.31 million in 2021).

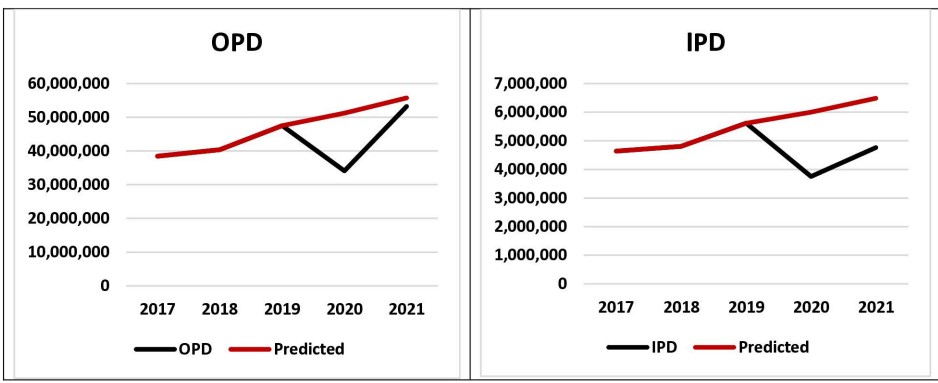

**Fig 1. Overall outpatient visits (OPD) and inpatient admissions (IPD).**

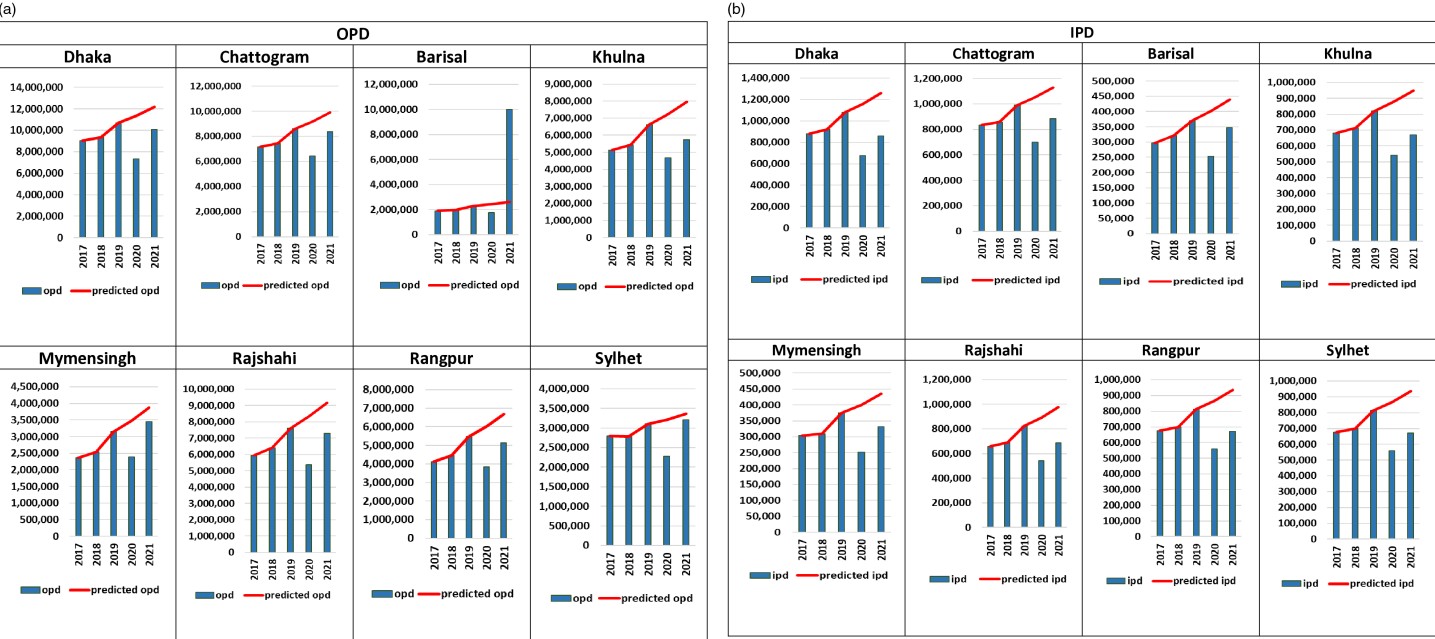

**Fig 2. (a) Annual changes in outpatient (OPDs) across divisions (2017–2021). (b) Annual changes in inpatients services (IPD) across divisions (2017–2021).**

### 3.4. Monthly change in case of OPD

Fig 3a shows monthly changes in OPD visits in divisional level for five years period (2017-2021). During pre COVID-19 period (2017-2019), we observed an increased number of OPD visits in October in all eight administrative divisions. The lowest OPD visits were seen during June (2017-2019) in Dhaka, Chattogram, Khulna, Rajshahi, Rangpur, and Sylhet divisions. The number of OPD visits drastically declined in April and May during 2020 in all divisions. In 2021, the number of OPD visits was highly increased in October (except for Rangpur division where OPD visits raised in September) which was similar to the pre COVID-19 period. In addition, lowest OPD visits were observed during May and July in 2021 for all the eight administrative divisions.

### 3.5. Monthly changes in IPD admissions

Fig 3b shows the number of monthly IPD admissions depending on months from 2017 to 2021. During pre-COVID-19 period (2017-2019), the highest number of IPD admissions observed in October but an exception was also seen for Barisal (April), Khulna (September) and Rajshahi (August) divisions. The number of IPD admissions were drastically dropped in all divisions during April and May in 2020. Highest number of the IPD admissions in all eight divisions were marked during January and February, in earlier 2020. The number of monthly IPD admissions showed an increasing but fluctuating trend in 2021 for all divisions. Moreover, the lowest IPD admissions were noticed in the month of July in 2021 for all the eight administrative divisions.

### 3.6. Segmented regression: OPD visit

The segmented regression results in Table 1 for outpatient department (OPD) visits across various sites in Bangladesh for 2020 and 2021 present changes in incident rate ratios (IRRs)

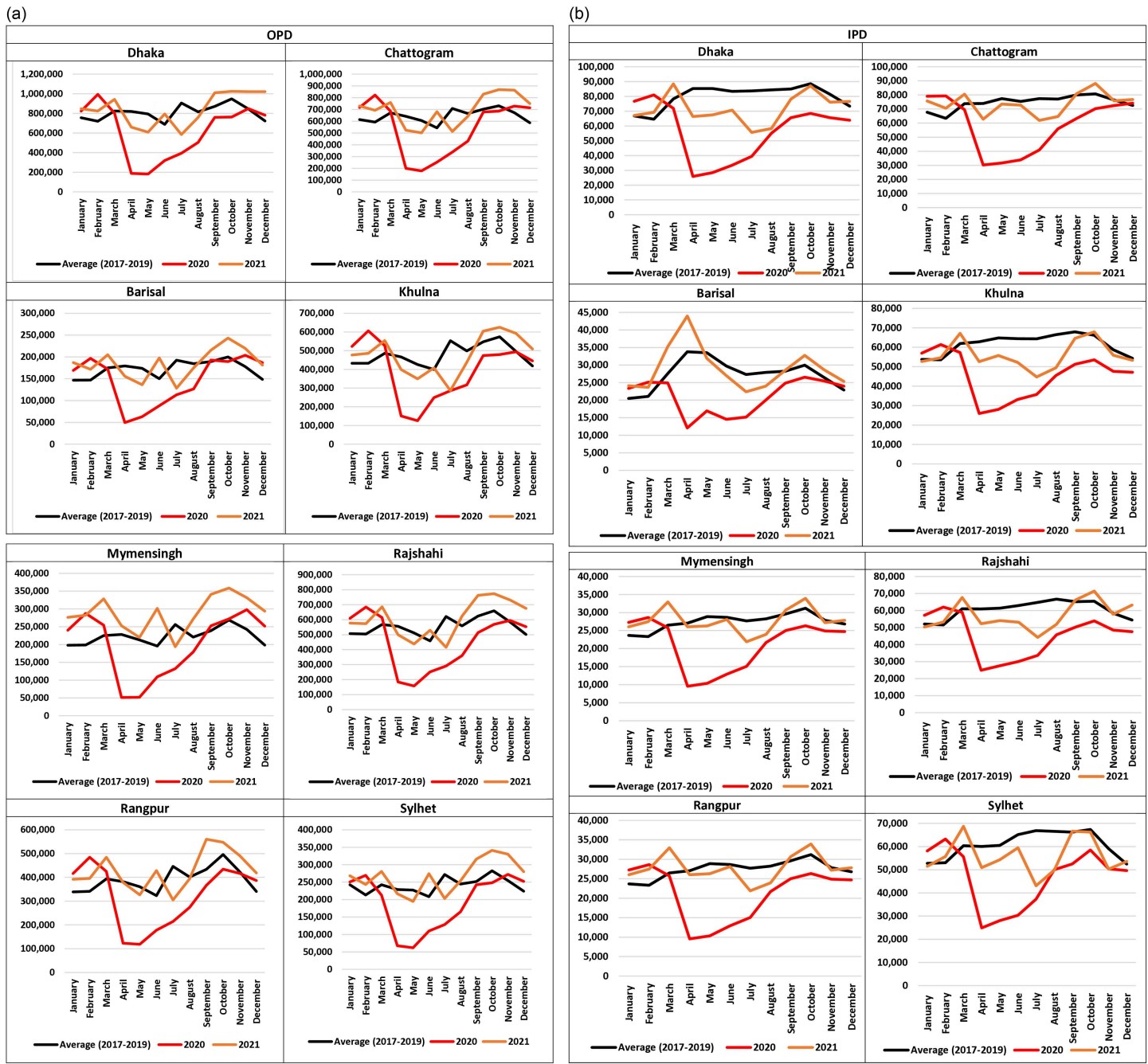

**Fig 3. (a) Monthly changes in outpatient visits (OPDs) across Bangladesh divisions (2017–2021). (b) Monthly changes in inpatient services (IPD) across Bangladesh divisions (2017–2021).**

and their statistical significance. In 2020, the IRRs for OPD visits significantly decreased in Bangladesh (IRR = 0.74; 95% CI: 0.57-0.98, P = 0.033), and divisions including Dhaka (IRR = 0.62; 95% CI: 0.47-0.82, P = 0.001), Barisal (IRR = 0.69; 95% CI: 0.54-0.87, P = 0.002), and Sylhet (IRR = 0.69; 95% CI: 0.57-0.84, P < 0.001), Mymensingh (IRR = 0.72; 95% CI: 0.54-0.96, P = 0.026). However, in 2021, the IRRs for most sites approximated 1, indicating no significant change from the baseline.

**Table 1.  Segmented regression: OPD.**

| Site | 2020 | | 2021 | |
|---|---|---|---|---|
| | IRR (CI) | P-value | IRR (CI) | P-value |
| Bangladesh | 0.74 (0.57–0.98) | **0.033** | 1.01 (0.97–1.05) | 0.475 |
| Dhaka | 0.62 (0.47–0.82) | **0.001** | 1.00 (0.97–1.04) | 0.865 |
| Chattogram | 0.75 (0.56–1.00) | 0.049 | 1.02 (0.98–1.05) | 0.414 |
| Barisal | 0.69 (0.54–0.87) | **0.002** | 1.03 (1.00–1.06) | 0.870 |
| Khulna | 0.81 (0.64–1.04) | 0.104 | 1.00 (0.97–1.03) | 0.982 |
| Mymensingh | 0.72 (0.54–0.96) | **0.026** | 1.03 (1.00–1.07) | 0.077 |
| Rajshahi | 0.79 (0.60–1.03) | 0.078 | 1.00 (0.97–1.04) | 0.804 |
| Rangpur | 0.77 (0.59–1.00) | 0.055 | 1.00 (0.998–1.04) | 0.578 |
| Sylhet | 0.69 (0.57–0.84) | **0.000** | 1.00 (0.98–1.03) | 0.544 |

### 3.7.  Segmented regression: IPD

The segmented regression results in Table 2 for outpatient department (OPD) visits across various sites in Bangladesh show significant reductions in 2020, with IRRs less than 1 and P-values all equal to 0.000. Specifically, the IRRs in 2020 were 0.68 (95% CI: 0.57-0.80) for Bangladesh, 0.64 (95% CI: 0.53-0.78) for Dhaka, 0.69 (95% CI: 0.58-0.83) for Chattogram, 0.69 (95% CI: 0.60-0.81) for Barisal, 0.70 (95% CI: 0.61-0.81) for Khulna, 0.66 (95% CI: 0.56-0.78) for Mymensingh, 0.69 (95% CI: 0.59-0.81) for Rajshahi, 0.72 (95% CI: 0.61-0.84) for Rangpur, and 0.68 (95% CI: 0.56-0.82) for Sylhet. However, in 2021, the IRRs for most sites were close to 1, indicating no significant change from the baseline, with P-values greater than 0.05.

## 4.  Discussion

The analysis of OPD visits and IPD admissions in Bangladesh from 2017 to 2021 reveals significant trends that highlight the impact of the COVID-19 pandemic on healthcare utilisation. Our findings demonstrated that the overall number of OPD visits and IPD admissions likewise dropped significantly during 2020 compared to previous year. However, later both the OPD visits and IPD admissions increased in 2021 though remained comparatively low. The lowest rebound for OPD visits and IPD admissions was noticed in Sylhet division in 2021. The number of OPD visits was drastically declined in April and May during 2020 in all divisions. The number of OPD visits was highest in October and lowest during May and July in 2021 for

**Table 2.  Segmented regression: IPD.**

| Site | 2020 | | 2021 | |
|---|---|---|---|---|
| | IRR (CI) | P-value | IRR (CI) | P-value |
| Bangladesh | 0.68 (0.57–0.80) | **0.000** | 0.99 (0.97-1.01) | 0.309 |
| Dhaka | 0.64 (0.53–0.78) | **0.000** | 0.98 (0.96-1.01) | 0.232 |
| Chattogram | 0.69 (0.58–0.83) | **0.000** | 0.99 (0.97-1.02) | 0.607 |
| Barisal | 0.69 (0.60–0.81) | **0.000** | 1.02 (1.00-1.03) | 0.075 |
| Khulna | 0.70 (0.61–0.81) | **0.000** | 0.98 (0.96-1.00) | 0.068 |
| Mymensingh | 0.66 (0.56–0.78) | **0.000** | 0.99 (0.97-1.01) | 0.464 |
| Rajshahi | 0.69 (0.59–0.81) | **0.000** | 0.98 (0.96-1.00) | 0.050 |
| Rangpur | 0.72 (0.61–0.84) | **0.000** | 0.99 (0.97-1.00) | 0.164 |
| Sylhet | 0.68 (0.56–0.82) | **0.000** | 1.00 (0.97-1.02) | 0.828 |

all the eight administrative divisions. The number of IPD admissions were drastically dropped in all divisions during April and May in 2020. Monthly IPD admissions showed an increasing but fluctuating trend in 2021 for all divisions.

The COVID-19 pandemic not only affected the healthcare system in Bangladesh, it also severely disrupted healthcare delivery in other neighbouring countries including India, Nepal, and Korea [20–22]. In 2020, OPD visits and IPD admissions in Bangladesh decreased significantly, with Dhaka experiencing the largest drop. The decline in IPD admissions was equally low, with admissions falling to 37.5 million. The segmented regression results show significant reductions in OPD visits across multiple sites. These significant decreases reflect the immediate impact of the pandemic. This drastic reduction of OPD visits and IPD admissions can be attributed to several factors, including lockdown measures [23], fear of contracting the virus, and the reallocation of healthcare resources to manage COVID-19 cases [24]. Besides, the healthcare system in Bangladesh is already fragile with limited resources and workforce [25]. Additionally, Bangladesh's health facilities also experienced limited outpatient visits during the COVID-19 pandemic [26]. Furthermore, the healthcare providers in Bangladesh were not immune to the virus, with some succumbing to COVID-19 and others getting infected in 2020 [27], which may have imposed an detrimental impact on the healthcare delivery in Bangladesh during the time.

In our study, the Outpatient Department (OPD) visit metrics revealed that Dhaka and Chattogram emerged as the most severely impacted divisions, which is consistent with other studies [28, 29]. Dhaka, recognised as the capital of Bangladesh, serves as a hub for administrative and various industrial activities. Chattogram, on the other hand, holds the status of being the commercial capital of the country. Together, these two divisions accommodate a staggering 60 million people, constituting the largest densely populated regions in Bangladesh. Furthermore, our findings indicate that Dhaka and Chattogram experienced the highest rates of COVID-19 infections and deaths [30]. The observed high infection and death rates underscore the challenges faced by densely populated regions and emphasise the need for targeted interventions and resource allocation to enhance healthcare capabilities in such critical areas.

In 2021, OPD visits rebounded across all divisions, meeting or exceeding the predicted values, where, the most significant rebound was observed in Barisal and lowest in Sylhet. The incidence rate ratios (IRRs) in this context indicating the relative change in the number of outpatient department (OPD) visits during the COVID-19 pandemic years (2020 and 2021) compared to the pre-COVID period (2017–2019). So, for most sites the IRR was close to pre-pandemic levels, indicating minimal change in healthcare utilisation. This suggested a stabilisation in healthcare services as the country adapted to the shifting pandemic situation. This recovery highlights the strength and ability of the healthcare system to adjust, driven by factors such as the government's initiation of a mass vaccination campaign in early 2021, which significantly reduced COVID-19 infections [31]. The combination of mass vaccination, the withdrawal of the nationwide lockdown, and the implementation of regional lockdowns likely played a key role in normalising outpatient (OPD) and inpatient (IPD) services [8]. Moreover, the rebound in 2021 for OPD visits and IPD admissions may also be attributed to a catch-up effect, as people who delayed or avoided seeking healthcare in 2020 due to lockdowns, fear of infection, or resource reallocation for COVID-19 cases, sought to address their accumulated health concerns once restrictions eased and vaccination efforts expanded.

The findings underscore the need for robust public health policies to manage healthcare disruptions during pandemics. Effective communication, infection control measures, and the reallocation of resources are crucial to maintaining healthcare services. The partial recovery observed in 2021 highlights the healthcare system's resilience but also indicates areas for improvement, particularly in ensuring sustained access to inpatient care. Further research is

needed to explore the long-term impacts of the pandemic on healthcare utilisation and outcomes. Studies focusing on the effectiveness of public health interventions, patient behaviour, and healthcare system resilience will provide valuable insights for managing future health crises.

The COVID-19 pandemic highlighted the vulnerability of healthcare systems, significantly hampering utilisation [32, 33]. To manage future pandemics, it is crucial to enhance public health infrastructure, ensure rapid and transparent communication, and maintain robust surveillance systems for early detection. Equitable vaccine distribution and adequate stockpiling of medical supplies will be essential to mitigate impacts [34]. Investing in research and fostering global cooperation will further strengthen pandemic preparedness [35].

## 5. Limitations

According to a survey conducted by Canada's Centre for International Epidemiological Training (CIET), 13% of treatment-seekers in Bangladesh receive government facilities, 27% receive private or non-profit services, and 60% receive unqualified services [36]. However, this study utilised data from public health facilities as reported in DHIS2 and thus could therefore not demonstrate the effect of COVID-19 on private healthcare institutions within Bangladesh. This implies limitation as the analysis does not include of patients utilising unqualified or private healthcare services in this study. The cross-sectional design poses a limitation in elucidating specific reasons for the decline in patient flow during the COVID-19 period, and a longitudinal approach might have provided more nuanced insights. Additionally, concerns about the quality of DHIS2 data, marked by incompleteness and inconsistency, were not addressed through data cleaning processes, introducing potential uncertainties, and affecting the precision of the analysis. The study is constrained by the absence of specific demographic data, such as age and gender, in the original dataset and lacks detailed information on hospital departmental units. The study's aggregated analysis, incorporating patient flow data from both inpatient and outpatient departments across all levels of healthcare facilities, lacks granularity, hindering the ability to report the specific impact on individual departments or gender preferences. Furthermore, the study does not specify which levels of healthcare facilities were most severely affected by the COVID-19 pandemic, limiting insights into the vulnerabilities and challenges faced by different tiers of healthcare institutions during this period.

## 6. Conclusions

The COVID-19 pandemic significantly reduced OPD visits and IPD admissions in Bangladesh in 2020, with partial recovery in 2021. This highlights the critical need to strengthen healthcare facilities and equip healthcare providers to be prepared for future pandemics or emergencies. Targeted interventions and resource allocation are crucial, particularly in densely populated regions which were severely impacted by the pandemic. Strengthening healthcare infrastructure, staffing, and services in these areas will better prepare them for future health crises. Further research is required to explore the long-term impacts of COVID-19 on the healthcare system and the capacity of the healthcare system to adapt and recover to face future emergencies and pandemics.

## Acknowledgments

Authors thank the competent authority and acknowledge the Directorate General of Health Services (DGHS) and the Ministry of Health and Family Welfare (MOHFW) of Bangladesh for providing access to the DHIS2 database and for the data to analyse in this study.

## Author contributions

**Conceptualization:** Aniqa Tasnim Hossain.

**Data curation:** Nasimul Ghani Usmani.

**Formal analysis:** K M Tanvir, Ema Akter.

**Methodology:** Ridwana Maher Manna, Md. Hafizur Rahman, Aniqa Tasnim Hossain.

**Resources:** M Sabbir Haider.

**Supervision:** Aniqa Tasnim Hossain.

**Writing – original draft:** Ridwana Maher Manna, Tasnu Ara, Nasimul Ghani Usmani.

**Writing – review & editing:** Md. Hafizur Rahman, M Sabbir Haider, Mohammad Sohel Shomik, Aniqa Tasnim Hossain.

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
