## [Decision Letter · Decision Letter 0]

4 Oct 2024

PONE-D-24-34344Impact of COVID-19 on the utilisation of in-patient and out-patient healthcare services in Bangladesh: analysis of routine healthcare system dataPLOS ONE

Dear Dr. Manna,

Thank you for submitting your manuscript to PLOS ONE. After careful consideration, we feel that it has merit but does not fully meet PLOS ONE’s publication criteria as it currently stands. Therefore, we invite you to submit a revised version of the manuscript that addresses the points raised during the review process.

We look forward to receiving your revised manuscript.

Kind regards,

Rajib Chowdhury, M.Sc.; MPH

Academic Editor

PLOS ONE

Journal Requirements:

2. Thank you for stating the following in your Competing Interests section: None 

3. In the online submission form, you indicated that data for this analysis were sourced from the DHIS2 Bangladesh database, dedicated to health-related indicators for routine facility monitoring and evaluation of public healthcare facilities. This data is owned by the government of Bangladesh and only available with proper request to the government.. 

Reviewers' comments:

Reviewer's Responses to Questions

**Comments to the Author**

1. Is the manuscript technically sound, and do the data support the conclusions?

Reviewer #1: Partly

Reviewer #2: Yes

2. Has the statistical analysis been performed appropriately and rigorously? 

Reviewer #1: I Don't Know

Reviewer #2: Yes

3. Have the authors made all data underlying the findings in their manuscript fully available?

Reviewer #1: Yes

Reviewer #2: No

4. Is the manuscript presented in an intelligible fashion and written in standard English?

Reviewer #1: No

Reviewer #2: Yes

5. Review Comments to the Author

Reviewer #1: 1. Title: The title is quite long. A more concise version would be: “Impact of COVID-19 on Healthcare Utilization in Bangladesh: Analysis of Routine Data.”

2. Short Title: Please correct the short title to: “Impact of COVID-19 on In-Patient and Out-Patient Care.”

3. Language: The article needs to be revised by a native English writer to improve readability and clarity.

Abstract:

4. Abbreviations: Abbreviations should be defined the first time they are used in the manuscript.

Introduction:

5. References: Provide the reference for lines 42-45.

6. References: Provide the reference for lines 70-72.

7. Clarity: The introduction needs to be rewritten to avoid repetition and to be more concise.

Methods:

8. Could you please provide a detailed paragraph outlining the assessment methods utilized in your research? This should include a description of both the direct and indirect methods employed, such as tests, surveys, interviews, or observational techniques, and how these methods contributed to the reliability and validity of your findings.

Data Analysis:

9. Using the Durbin-Watson statistic to check for autocorrelation is correct. However, note that the Durbin-Watson test is typically used for linear regression models. For Poisson regression, you might consider using other methods like the autocorrelation function (ACF) or partial autocorrelation function (PACF). (line 135)

10. Clarify whether you used Poisson regression or linear regression for the final analysis. If Poisson regression was used, remove the mention of applying a linear regression model. (line 137)

Results:

11. Please correct the sentence in lines 147-148 to: “The number of OPD visits fell in 2020 (34.1 million) compared to 2019 but showed an upward trend, reaching the targeted predicted value in 2021.”

12. Correct “increases” to “increased” in line 150.

13. Line 220, table 2: The confidence interval for Dhaka in 2020 is listed as (0.83-0.78), which seems incorrect because the lower bound should be less than the upper bound.

Discussion:

14. Please compare your major findings to previous studies found in the literature.

15. Write a small paragraph recommending possible measures to manage such pandemics in the future, while citing experiences from previous pandemics

Reviewer #2: Overall this is an interesting piece of work to validate the hypothesis that the COVID-19 pandemic impacted utilisation of outpatient and inpatient care.

-The introduction is very well written and well describes the context of the study and need to better understand the problem.

The materials and methods

-Generally are well written and clear. PlosOne required the data to be made publicly available so please recheck this criteria/allowances.

-Indicators on line 117: would be nice to highlight if certain departments were not captured in the DHIS

Results

-Lines147-151: Suggest a minor rewriting as at the moment not clear if difference in visits were 34.1million and 37.5million or declined to that value.

-Line 158: Results should not traditionally contain any subjective statements e.g. "interesting". Would rather suggest a rephrasing to state exactly what the trend was (i.e. sudden increase etc.) as a simple statement.

-Line 166-67: Again, please re-highlight this point in the discussion offering some deductions/explanations.

-Line 167-8: Please double check this grammar through the document as should state "number of IPD admissions declined" and remove the word "were" from this statement. In Line 168 please also remove the word substantial and replace with something more objective.

-Line 170-71: Suggest replacing "could not touch" to "did not reach the predicted values"

-Line 179: remove the word "was" from "OPD visits was drastically declined". Likewise in line 180.

-Line 188: See suggestion about from lines 167-8

Discussion

-Line 260-261: Would be nice to offer some suggestion as to why this significant rebound was observed, do we think it's due to catch-up from backlog?

Limitations:

-Line 285: Please state what proportion of the population access healthcare in private facilities to understand what amount of data is missing from this analysis. Important to understand how we can apply the results to the general population of Bangladesh.

-Line 290-93: Please explain why this more detailed analysis on age/departments/gender could not be assessed. Is that due to limitations with data access, limitations in the data itself or not the purpose of this study.

Figures:

-Minor suggestion to include commas in the y-axis to help readability of the numbers

6. PLOS authors have the option to publish the peer review history of their article (what does this mean? ). If published, this will include your full peer review and any attached files.

**Do you want your identity to be public for this peer review?** For information about this choice, including consent withdrawal, please see our Privacy Policy .

Reviewer #1: **Yes: ** Batoul Ghosn, PhD

Reviewer #2: No

---

## [Author Response · Author response to Decision Letter 1]

14 Nov 2024

Editors/Journal requirements:

*1. We note that your Data Availability statement states the following: "Data for this analysis were sourced from the DHIS2 Bangladesh database, dedicated to health-related indicators for routine facility monitoring and evaluation of public healthcare facilities. This data is owned by the government of Bangladesh and only available with proper request to the government."

If the data is only available upon request, please provide contact information, such as an email address, for a non-author, institutional point of contact (such as an IRB or ethics committee contact) who can field data inquiries from fellow researchers. If the data contact is an individual, please provide their title and relationship to the data as well. Please note that PLOS does not allow authors to be the sole contact for data inquiries.

Response: Thank you so much for your instant feedback. Data for this analysis were sourced from the DHIS2 Bangladesh database, dedicated to health-related indicators for routine facility monitoring and evaluation of public healthcare facilities. This data is owned by the government of Bangladesh and only available with proper request to the government. For data inquiries, please contact Dr. Md Toufiq Hassan Shawon, Medical Officer at the Management Information System (MIS), Directorate General of Health Services (DGHS), at sawontheboss4@gmail.com. Dr. Shawon is responsible for managing and overseeing data access for this study.

We have also updated the information in the answer box in the data availability section.

Response: Thank you for your feedback. We have addressed the comment according to the suggestion.

2. Thank you for stating the following in your Competing Interests section: None

Response: Thank you for the comment. We have updated the cover letter and mentioned about the competing interest in the cover letter as "The authors have declared that no competing interests exist."

3. In the online submission form, you indicated that data for this analysis were sourced from the DHIS2 Bangladesh database, dedicated to health-related indicators for routine facility monitoring and evaluation of public healthcare facilities. This data is owned by the government of Bangladesh and only available with proper request to the government..

Response: Thank you for your comment regarding DHIS2 data availability for more clearance. We received this data from the Director General of Health Services of Bangladesh Government, who owns the data, and we do have permission to make the data publicly available or share data with manuscript. Besides, the DHIS2 is owned by the Government of Bangladesh which is available on the DHIS2 website (https://centraldhis.mohfw.gov.bd/dhismohfw/dhis-web-commons/security/login.action). Anyone can access the data upon issuing an email request to the Director General of Health Services of Bangladesh Government.

Response: Thank you for your feedback.

Reviewer #1:

1. Title: The title is quite long. A more concise version would be: “Impact of COVID-19 on Healthcare Utilization in Bangladesh: Analysis of Routine Data.”

Response: Thank you for your feedback. We appreciate your insights about the title. We concise the title according to your suggestion into “Impact of COVID-19 on In-Patient and Out-Patient services in Bangladesh”.

2. Short Title: Please correct the short title to: “Impact of COVID-19 on In-Patient and Out-Patient Care.”

Response: Thank you for your insightful observations. We corrected the short title according to the suggestion.

3. Language: The article needs to be revised by a native English writer to improve readability and clarity.

Response: Thank you for your insightful observations. We made a significant revision to the narrative of the manuscript with help of a native English writer.

Abstract:

4. Abbreviations: Abbreviations should be defined the first time they are used in the manuscript.

Response: Thank you for your insightful observations. We have addressed the comment regarding abbreviations in the abstract section.

Introduction:

5. References: Provide the reference for lines 42-45.

Response: Thank you for your feedback. We have updated the references and introduction section of the manuscript.

6. References: Provide the reference for lines 70-72.

Response: Thank you for your observations. We have updated the introduction section of the manuscript and added the references.

7. Clarity: The introduction needs to be rewritten to avoid repetition and to be more concise.

Response: Thank you for your comment. We have revised the introduction accordingly in a concise way.

Methods:

8. Could you please provide a detailed paragraph outlining the assessment methods utilized in your research? This should include a description of both the direct and indirect methods employed, such as tests, surveys, interviews, or observational techniques, and how these methods contributed to the reliability and validity of your findings.

Response: Thank you for your insightful comments. We have revised the paragraph as suggested. Our analysis primarily included data from District Health Information System (DHIS2) which aggregates healthcare utilization data from public hospitals in Bangladesh. By design, it’s a secondary analysis of health system data. To enhance the readability and clarity of the assessment method utilized during the generation and management of the data, we will include detailed explanation of how these data were recorded at the health facility, validated, and deposited into the DHIS2 system. We will also describe the data flow from the health facilities to the national database, as well as the storage and quality control process. Additionally, no direct method such as surveys or interviews were used since our study relied on existing data. DHIS2 ensures data accuracy through routine check and validation procedures at various administration levels, which contribute to the reliability of our findings.

The revised paragraph is provided below and we have added the paragraph according to the suggestion in line number 79-97 highlighted in the manuscript.

“The data used in this study were sourced from the District Health Information System, version 2 (DHIS2), which collects aggregated data from public healthcare facilities in Bangladesh (14). The data included inpatient admission and outpatient visits from all 64 district hospitals across all eight admirative division of Bangladesh (15). Health facilities record patient data at the facilities using paper-based or electronic systems. These data are then digitalized and uploaded to the DHIS2 platform by a designated facility staff in every facility. The uploaded data flows from the health facilities to district health offices, where it is reviewed, validated, and then sent to the national level (14).

DHIS2 employs a robust data validation process, including routine data check, cross-verification, and quality control in different administrative levels to ensure the accuracy and completeness of the data (14). Although no direct methods such as surveys or interviews were employed in this study, the data was considered reliable due to the rigorous process DHIS2 implement for the quality assurance (16). These processes include standardized data entry protocols at the facility level, where healthcare staff input data on outpatient visits and inpatient admissions using electronic or paper-based forms (17). After entry, data undergoes multiple levels of review, including validation checks at district health offices, where errors or inconsistencies are flagged and corrected. Additionally, the DHIS2 platform perform automated data quality checks to ensure completeness and consistency before the data is consolidated at the national level (18).”

Data Analysis:

9. Using the Durbin-Watson statistic to check for autocorrelation is correct. However, note that the Durbin-Watson test is typically used for linear regression models. For Poisson regression, you might consider using other methods like the autocorrelation function (ACF) or partial autocorrelation function (PACF). (line 135)

Response: Thank you for your insightful observations. We addressed the comment according to the suggestion in the “Data Analysis” section under the Materials and Methods section (Line 118-120).

10. Clarify whether you used Poisson regression or linear regression for the final analysis. If Poisson regression was used, remove the mention of applying a linear regression model. (line 137)

Response: We appreciate your feedback regarding the materials and methods section of the article. We have mentioned the Poisson regression model for this study (Line 113 to 116).

Results:

11. Please correct the sentence in lines 147-148 to: “The number of OPD visits fell in 2020 (34.1 million) compared to 2019 but showed an upward trend, reaching the targeted predicted value in 2021.”

Response: Thank you for your insightful observations. We addressed the comment according to the suggestion in the Result section (Line 125-126).

12. Correct “increases” to “increased” in line 150.

Response: Thank you for your feedback. We updated the word according to the suggestion (Line 128).

13. Line 220, table 2: The confidence interval for Dhaka in 2020 is listed as (0.83-0.78), which seems incorrect because the lower bound should be less than the upper bound.

Response: Thank you for your comment. We addressed the comment according to the suggestion in the Result section (Line 188 and table 2).

Discussion:

14. Please compare your major findings to previous studies found in the literature.

Response: Thanks for your valuable suggestion. We have revised the discussion portion mentioning the major findings comparing with previous literature as per your suggestion.

15. Write a small paragraph recommending possible measures to manage such pandemics in the future, while citing experiences from previous pandemics

Response: Thank you for your feedback. We appreciate your insights about the management of future pandemic. We have included a paragraph in the end of Discussion section (Line 262-267).

Reviewer #2: Overall, this is an interesting piece of work to validate the hypothesis that the COVID-19 pandemic impacted utilisation of outpatient and inpatient care

Response: Thank you for your appreciation

-The introduction is very well written and well describes the context of the study and need to better understand the problem.

Response: Thanks for the valuable comment and appreciation. We have revised the introduction section and shed more light on the problem.

The materials and methods

-Indicators on line 117: would be nice to highlight if certain departments were not captured in the DHIS

Response: We appreciate this valuable suggestion. We acknowledge that DHIS2 captures data primarily from public healthcare facilities and does not include private sector data. Moreover, the aggregated nature of the data limits the granularity, meaning that department-specific healthcare utilization could not be analyzed (Line 100-106). We will include this limitation in the manuscript.

Results

-Lines147-151: Suggest a minor rewriting as at the moment not clear if difference in visits were 34.1million and 37.5million or declined to that value.

Response: Thank you for your insightful observations. We have addressed the comment according to the suggestion in the Result section (Line 125-129).

-Line 158: Results should not traditionally contain any subjective statements e.g. "interesting". Would rather suggest a rephrasing to state exactly what the trend was (i.e. sudden increase etc.) as a simple statement.

Response: Thank you for your precise feedback. We updated the wordings according to your suggestion (Line 135-136).

-Line 166-67: Again, please re-highlight this point in the discussion offering some deductions/explanations.

Response: We appreciate this valuable suggestion. According to your suggestion we have revised and added appropriate information to the discussion section of the manuscript.

-Line 167-8: Please double check this grammar through the document as should state "number of IPD admissions declined" and remove the word "were" from this statement. In Line 168 please also remove the word substantial and replace with something more objective.

Response: We appreciate this valuable suggestion. We have addressed the comment according to the suggestion (Line145-147).

-Line 170-71: Suggest replacing "could not touch" to "did not reach the predicted values"

Response: Thank you for your feedback. We updated the word according to the suggestion (Line 147-149).

-Line 179: remove the word "was" from "OPD visits was drastically declined". Likewise in line 180.

Response: We appreciate your feedback regarding the result section of the article. We addressed the comment in (Line 155 to 157) the result section of the manuscript.

-Line 188: See suggestion about from lines 167-8

Response: Thank you for your insightful observations. We addressed the comment according to the suggestion in the Result section (Line 164-166).

Discussion

-Line 260-261: Would be nice to offer some suggestion as to why this significant rebound was observed, do we think it's due to catch-up from backlog?

Response: Thank you for your insightful observations. We addressed the comment according to the suggestion in the discussion section (Line 248-252)

Limitations:

-Line 285: Please state what proportion of the population access healthcare in private facilities to understand what amount of data is missing from this analysis. Important to understand how we can apply the results to the general population of Bangladesh.

Response: We appreciate this valuable suggestion. We acknowledge that DHIS2 captures data primarily from public healthcare facilities and does not include private sector data. Which we included in the limitation in the manuscript (Line 269-275).

-Line 290-93: Please explain why this more detailed analysis on age/departments/gender could not be assessed. Is that due to limitations with data access, limitations in the data itself or not the purpose of this study.

Response: Thank you for the comment. You rightly mentioned that inclusion of age/gender could provide better understanding of the findings. Additionally, this is facility level aggregated data which does not report the individual level data that can provide the information on variables like gender. However, other departments including the inpatient and outpatient are there, although, it was not possible to further segregate into departments such as pediatr

---

## [Editor Report · Decision Letter 1]

28 Nov 2024

Impact of COVID-19 on In-Patient and Out-Patient services in Bangladesh

PONE-D-24-34344R1

Dear Dr. Manna,

We’re pleased to inform you that your manuscript has been judged scientifically suitable for publication and will be formally accepted for publication once it meets all outstanding technical requirements.

Kind regards,

Rajib Chowdhury, M.Sc.; MPH

Academic Editor

PLOS ONE
---

## [Editor Report · Acceptance letter]

PONE-D-24-34344R1

PLOS ONE

Dear Dr. Manna,

I'm pleased to inform you that your manuscript has been deemed suitable for publication in PLOS ONE. Congratulations! Your manuscript is now being handed over to our production team.

Kind regards,

on behalf of

Dr. Rajib Chowdhury

Academic Editor

PLOS ONE